# Missed opportunities in hypertension risk factors screening in Indonesia: a mixed-methods evaluation of integrated health post (POSBINDU) implementation

Vitri Widyaningsih [iD],[1] Ratih Puspita Febrinasari,[2] Eti Poncorini Pamungkasari,[1] Yusuf Ari Mashuri,[3] Sumardiyono Sumardiyono,[1] Balgis Balgis,[1] Jaap Koot,[4] Jeanet Landsman-Dijkstra,[4] Ari Probandari [iD],[1] on behalf of Scaling Up Non-Communicable Disease Intervention in South East Asia (SUNISEA) Project

[1]Public Health, Universitas Sebelas Maret, Surakarta, Indonesia
[2]Pharmacology, Universitas Sebelas Maret, Surakarta, Indonesia
[3]Parasitology, Universitas Sebelas Maret, Surakarta, Indonesia
[4]Department of Health Sciences, University Medical Centre Groningen, Groningen, The Netherlands

**Correspondence to**
Dr Ari Probandari;
ari.probandari@staff.uns.ac.id

## ABSTRACT

**Objectives** To assess the implementation and contextual barriers of POSBINDU, a community-based activity focusing on screening of non-communicable diseases (NCDs), mainly hypertension and diabetes, in Indonesia.

**Design** This was a concurrent mixed-methods study, with a cross-sectional analysis of secondary data and focus group discussions (FGDs) on stakeholder of POSBINDU.

**Setting** The study was conducted in seven districts in three provinces in Indonesia, with approximately 50% of the primary healthcare (PHC) were selected as areas for data collection (n PHC=100).

**Participants** From 475 POSBINDU sites, we collected secondary data from 54 224 participants. For the qualitative approach, 21 FGDs and 2 in-depth interviews were held among a total of 223 informants.

**Primary outcomes and measures** Proportion of POSBINDU visitors getting the hypertension screening and risk factors' assessment, and barriers of POSBINDU implementation.

**Results** Out of the 114 581 POSBINDU visits by 54 224 participants, most (80%) were women and adults over 50 years old (50%) showing a suboptimal coverage of men and younger adults. Approximately 95.1% of visitors got their blood pressure measured during their first visit; 35.3% of whom had elevated blood pressure. Less than 25% of the visitors reported to be interviewed for NCDs risk factors during their first visit, less than 80% had anthropometric measurements and less than 15% had blood cholesterol examinations. We revealed lack of resources and limited time to perform the complexities of activities and reporting as main barrier for effective hypertension screening in Indonesia.

**Conclusions** This study showed missed opportunities in hypertension risk factors screening in Indonesia. The barriers include a lack of access and implementation barriers (capability, resources and protocols).

## INTRODUCTION

The increasing trends of non-communicable diseases (NCDs) in the world, including Indonesia, require targeted and specific primary and secondary prevention.[1][2] Hypertension, one of the most common NCDs, has a

## Summary

### Findings
► In a mixed-methods study, we found suboptimal implementation of POSBINDU which reflected the missed opportunities in screening for hypertension and its risk factors in Indonesia. Several barriers include suboptimal coverage, complexities of activities and overlap between different non-communicable disease-related programmes, and lack of resources.

### Implications
► There is a need to improve coverage and implementation of POSBINDU for screening for hypertension and its risk factors. An integrated approach to improve the implementation of hypertension screening from guidelines to practice is crucial.

### Strengths and limitations of this study
► This was a relatively large evaluation of POSBINDU in Indonesia, with almost 2 years of data.
► The findings from mixed-methods study provide more comprehensive information on POSBINDU implementation.
► Information on the contextual factors of POSBINDU implementation can provide insights into steps to improve POSBINDU in the communities.
► The use of secondary data poses variations in blood pressure and anthropometrics measurements.
► The study limitation also includes the difficulty in differentiating whether the missed reporting was due to lack of activities or lack of reporting. Nevertheless, both the activities and reporting are important in non-communicable diseases screening, particularly in the follow-up.

relatively high (33.4%) prevalence in Indonesia.[3][4] This figure is estimated to increase even further with the changing (more sedentary) lifestyle, unhealthy diet, rising prevalence of obesity and the increasing life expectancy.[5] In 2015, hypertension attributed to 41% of all disability-adjusted life-years lost,

and was the leading risk factor for cardiovascular diseases.[6] Economically, hypertension accounts for $370 billion in medical costs per year worldwide.[7] Major modifiable risk factors for NCDs include smoking, alcohol consumption, unhealthy diet and obesity, and a sedentary lifestyle.[8 9] With the heavy burden and the economic cost of this disease, primary and secondary prevention for hypertension and its risk factors become very important.

In 2010, the WHO has recommended the implementation of Package of Essential Interventions for Non-Communicable (PEN) Diseases for low/middle-income countries.[10] In response, the Ministry of Health (MOH) in Indonesia launched the Integrated Health Post (POSBINDU), as part of the PEN programme. POSBINDU, a community-based programme for hypertension screening and prevention,[11] was added to the several existing NCD-related programmes of Indonesia. These include Prolanis (Program Pengendalian Penyakit Kronis), a community-based hypertension and diabetes management programme affiliated with primary care[12] and Posyandu Lansia, a community-based NCDs screening and management for the elderly.[13] Despite these efforts, the awareness and control of hypertension are still relatively low: only 25% of people with elevated blood pressure are aware of their condition, and only 54% of people diagnosed with hypertension take routine medication.[4 14 15] These conditions are still below the 'rule of halves' for hypertension management, which recommends that 50% of hypertension patients be aware of their condition, with half of whom should be treated.[16 17]

A process evaluation is important in assessing the implementation, to identify barriers, and provide specific recommendations for improvement of POSBINDU. Previous studies have evaluated the effectiveness of the POSBINDU implementation.[11 18] However, they were lacking on the evaluation of contextual barriers in POSBINDU implementation. This study aims to portray the implementation of POSBINDU and its contextual barriers, to provide recommendations for better hypertension and its risk factors screening, and optimal linkage to care in Indonesia.

## METHODS
### Setting
POSBINDU is a community-based activity run by community health cadres (volunteers) and supervised by primary health care (PHC) officials. POSBINDU aims to empower communities in screening for NCDs and the risk factors, targeting individuals above 15 years old, particularly those of productive age.[19 20] The main activities include screening for NCDs (mainly hypertension and diabetes) and the risk factors (ie, smoking, diet, physical activity, obesity). Further, POSBINDU also provides health education and facilitates referral to PHC.[19] For this study, we focus on POSBINDU implementation in screening of hypertension and its risk factor, particularly, since only 30% of hypertensive patients in Indonesia received formal diagnosis.[15]

### Study design
This was a concurrent mixed-methods study in seven districts in three provinces in Indonesia (Central Java, East Java and North Sumatra). We purposely selected provinces with relatively high prevalence of NCDs based on a national health survey conducted in 2018.[21] Cross-sectional study by obtaining POSBINDU reports was conducted for the quantitative evaluation, whereas case study was conducted to explore barriers of POSBINDU implementation.

### Data collection
Within every one of the three provinces, we selected two districts: one city representing urban communities, and one district representing rural communities. In Central Java, an additional city was also selected (figure 1). The rural/urban classification is based on population density and facilities available in the communities. For each district, approximately

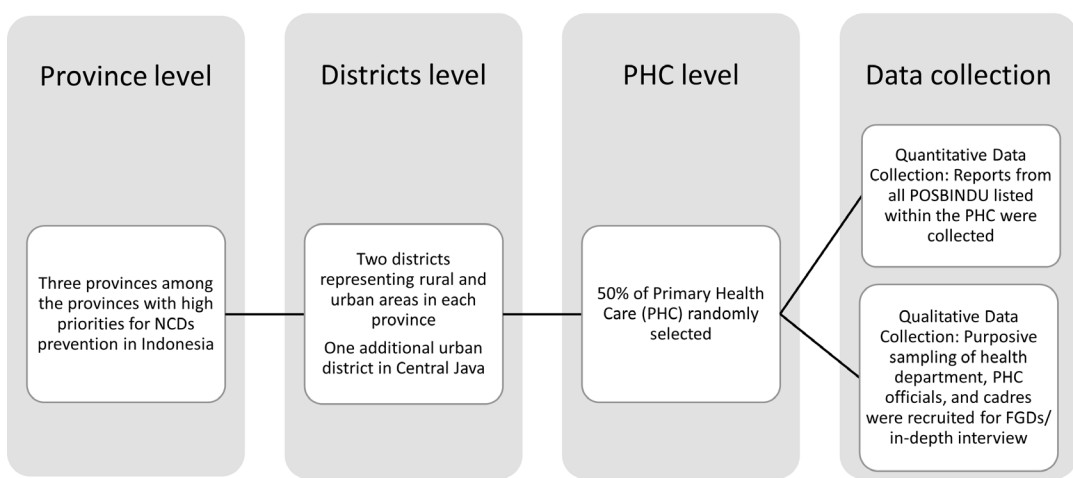

**Figure 1** Study sample selection. FGDs, focus group discussions; NCD, non-communicable disease.

50% of the PHC were selected as areas for data collection (n PHC=100). Within the PHC, we collected data for quantitative process evaluation from all active POSBINDU in the areas (n POSBINDU=475). Due to the different number of POSBINDU within each district or PHCs, the number of POSBINDU visitors as well as visits varies by the areas. In most POSBINDU, online/electronic data were not available. Hence, data on participation were manually collected from the POSBINDU register. Data from 2018 to 2019 were collected, except for Central Java, in which data were available through September 2019.

For the qualitative approach, 21 focus group discussions (FGDs) and 2 in-depth interviews were held among a total of 223 informants: 22 from Districts Health Department, 101 from PHC facilities and 100 POSBINDU cadres. The two in-depth interviews were conducted with health districts department officials. Within each district, we conducted purposive sampling to recruit health officials responsible for POSBINDU programme from the district's health department, and PHC. We also recruit 2–3 cadres from each PHC based on list of cadres obtained from PHC officials. These participants were recruited to obtain information on POSBINDU implementation facilitators and barriers. The size of the FGDs was on average 10 persons (min 4, max 18). Verbatim transcripts of the FGD's were made for qualitative analyses. The FGD facilitators had public health background and experience in conducting qualitative research. All facilitators attended the preparatory meeting to discuss the FGDs and interview guidelines, to obtain similar perception regarding the aims of FGDs and interviews and items of the FGD guidelines.

### Outcome and variables measurements

Missed opportunities in hypertension screening were quantified by the proportion of POSBINDU visitors getting the risk factors anamnesis, and measurement of anthropometric tests, blood pressure and cholesterol. Analyses were conducted on each indicator to provide more detailed information on specific components of screening which were lacking. Sociodemographic variables which were available on the POSBINDU register, were included in the analyses: sex, age and level of education. Age was classified into several groups based the Indonesian Ministry of Health classification for age (youth=15–24 years old, adult=25–44 years old, pre-elderly=45–59 years old and elderly=>60 years old). Occupation was not included in the analyses due to high missing value in the POSBINDU reports (>60%).

Personal and family history of NCDs were also obtained, which include seven diseases: hypertension, diabetes, heart disease, stroke, asthma, cancer and high blood cholesterol. Complete personal/family history variables were coded 1 if all information was available and coded 0 if at least one of the disease histories was missing. Any personal/family history variables were coded 1 if at least one of the disease histories was available and coded 0 if all of the history information was missing.

We also generate variable 'incomplete information' which represents whether the individual received the recommended procedure (history taking, anthropometric measurement, blood pressure measurement and blood examination). The proportion presented in the analyses described the individuals who did not receive the complete recommended procedure.

We used the logic model framework for process evaluation to assess the implementation of POSBINDU. We adopted several indicators from the current literature on the use of logic model in process evaluation of community-based health intervention.[22–24] The FGDs theme as well as indicators of the secondary data developed based on the literature were discussed with officials from health department and PHC officials in one pilot site for finalisation. We further explored the barriers of POSBINDU implementation using a qualitative approach.

### Analyses

Statistical analyses were conducted using STATA, to calculate the proportion of activities and outcomes. We further conduct $\chi^2$, t-test and Analysis of Variance (ANOVA) to assess the statistical significance of the differences. Analyses were conducted on missing information, reflecting whether specific procedure in POSBINDU was carried out and reported. Further analyses on proportion of hypertension and body mass index status were also conducted. The two indicators were reported due to relatively high availability of these data (92% and 76%) compared with other indicators. Verbatim transcript from FGDs and in-depth interviews recordings were analysed. Content analysis was applied for the qualitative data to ascertain barriers for the POSBINDU implementation in Indonesia by two independent researchers. To enhance trustworthiness, we assess barriers of POSBINDU from several sources for triangulation purposes: health and PHC officials to reflect implementer's perspective, and cadres to reflect implementers and users' perspective. During data analyses, we also discuss the findings with representative of the FGD participants, that is, member checking. Parallel analyses were conducted to synthesise the findings from the quantitative and qualitative approaches. Weaving technique, analysing the quantitative and qualitative findings together by theme or concept, was used to integrate the findings.[25]

### Patients and public involvement

Patients or the public were not directly involved in the design, or conduct, or reporting, or dissemination plans of our research.

## RESULTS
### Participation of community for hypertension screening in POSBINDU

Data from 114 581 POSBINDU visits (54 224 participants) were analysed. The findings showed similar patterns in the districts and provinces: more female and elderly participants. Approximately 80% were female participants, with the highest proportion of female participants in rural North Sumatra (95.5%). Meanwhile, in Java, a

**Table 1** Characteristics of POSBINDU participants within the three provinces in Indonesia (POSBINDU register, 2018–2019)

| Characteristics | North Sumatra | | East Java | | Central Java | | Total |
|---|---|---|---|---|---|---|---|
| | **Rural** | **Urban** | **Rural** | **Urban** | **Rural** | **Urban** | |
| Number of individuals | 5103 | 10999 | 23053 | 4983 | 3398 | 6688 | 54224 |
| Number of PHC | 11 | 23 | 29 | 9 | 11 | 17 | 100 |
| Number of POSBINDU | 38 | 38 | 283 | 27 | 27 | 62 | 475 |
| *Categorical (%, SE)* | | | | | | | |
| Female | 95.5 (0.3) | 71.3 (0.4) | 76.2 (0.3) | 86.7 (0.5) | 73.7 (0.8) | 88.2 (0.4) | 79.4 (0.2) |
| Age | | | | | | | |
| 15–24 | 8.7 (0.4) | 6.1 (0.3) | 6.2 (0.2) | 7.8 (0.4) | 13.3 (0.7) | 3.8 (0.3) | 6.7 (0.1) |
| 25–44 | 38.1 (0.8) | 24.9 (0.5) | 33.0 (0.3) | 48.7 (0.8) | 47.7 (1.0) | 28.9 (0.7) | 22.2 (0.2) |
| 45–59 | 30.7 (0.7) | 37.6 (0.6) | 33.2 (0.3) | 31.0 (0.6) | 30.1 (0.9) | 42.4 (0.7) | 24.8 (0.2) |
| >60 | 22.4 (0.6) | 31.3 (0.6) | 27.6 (0.3) | 12.5 (0.5) | 8.9 (0.6) | 24.9 (0.6) | 24.7 (0.2) |
| Education | | | | | | | |
| PS | 2.6 (0.2) | 15.3 (0.3) | 57.3 (0.3) | 50.5 (0.7) | 41.1 (0.8) | 25.2 (0.5) | 38.0 (0.2) |
| HS | 0.3 (0.07) | 0.7 (0.1) | 0.3 (0.0) | 1.3 (0.2) | 2.3 (0.3) | 1.8 (0.2) | 0.8 (0.0) |
| Univ | 0.0 (0.0) | 2.7 (0.2) | 0.4 (0.04) | 4.9 (0.3) | 4.9 (0.4) | 2.1 (0.2) | 1.8 (0.1) |
| Missing | 97.1 (0.2) | 81.4 (0.4) | 41.9 (0.3) | 43.3 (0.7) | 51.8 (0.9) | 70.8 (0.6) | 59.4 (0.2) |
| Number of visits | | | | | | | |
| 1 time | 87.0 (0.4) | 77.4 (0.3) | 68.4 (0.3) | 65.6 (0.7) | 84.5 (0.6) | 56.4 (0.6) | 71.2 (0.1) |
| 2–6 times | 12.9 (0.5) | 21.7 (0.3) | 21.1 (0.2) | 23.9 (0.6) | 13.3 (0.6) | 35.1 (0.6) | 22.0 (0.1) |
| 7–12 times | 0.1 (0.0) | 0.6 (0.1) | 6.3 (0.2) | 6.0 (0.3) | 2.1 (0.2) | 5.9 (0.3) | 5.3 (0.1) |
| >12 times | 0 (0.0) | 0.2 (0.1) | 4.1 (0.1) | 4.5 (0.3) | 0.1 (0.0) | 2.6 (0.2) | 1.4 (0.1) |
| *Continuous (mean, SE)* | | | | | | | |
| Age | 46.4 (0.2) | 51.7 (0.2) | 49.7 (0.1) | 43.4 (0.2) | 41.0 (0.3) | 50.4 (0.2) | 48.6 (0.8) |
| Number of visits | 1.2 (0.8) | 1.4 (0.1) | 2.5 (0.2) | 2.7 (0.5) | 1.4 (0.2) | 2.5 (0.3) | 2.1 (0.1) |

Within province, rural–urban comparisons are significant at 0.05.
Between provinces, comparisons are significant at 0.05.
Differences in proportion tested using $\chi^2$.
HS, high school; Missing, data missing; PHC, primary healthcare; PS, primary school/less; Univ, university/college.

higher proportion of female participants were observed in urban areas (table 1).

Despite the relatively high missing information on age (n missing=12084, or 22.3%), we found that the participants were on average of older age, with roughly 50% of participants aged over 45 years old, almost 25% were >60 years old (table 1). The highest proportion of participants >60 years old was observed in rural East Java (31.3%), with mean age of 51.7 years old. We measured the youngest POSBINDU participants in rural Central Java (mean age 41.0 years old). Meanwhile, the missing information on education level was higher (almost 60%), with even higher proportion in North Sumatra.

In the span of the 2 years of secondary data collection, we found that, on average, the participants visit POSBINDU two times, with the lowest average of visits in North Sumatra (rounded to 1 visit/participant). Approximately 38628 (71.2%) of participants visit POSBINDU once for 2 years, and 761 (1.4%) visits POSBINDU more than 12 times.

We further observed the relatively high missing information for screening in POSBINDU across the districts, with the following general pattern. First, a relatively high proportion of missing information concerning the personal and family history, with East Java having the lowest proportion. Second, a relatively lower proportion of missing data on anthropometric measurements (less than 50%). Third, in all seven districts, the highest proportion of available data were for blood pressure measurements, followed by weight and height information. Last, our analysis identified higher missing values for blood cholesterol measurements (84,2%). For all measurements, there were significant differences between the three provinces, as well as between the rural and urban areas within the provinces (table 2).

Based on available data, we found that obesity seems to be more prevalent in urban areas in Java, but relatively similar between rural and urban areas in North Sumatra. In contrast, hypertension was more prevalent in a rural area for East Java and North Sumatra but was

**Table 2** Missing information and risk factors characteristics within POSBINDU participants (POSBINDU register, 2018–2019)

| | North Sumatra | | East Java | | Central Java | | |
| | Rural | Urban | Rural | Urban | Rural | Urban | Total |
| Characteristics | % (SE) | % (SE) | % (SE) | % (SE) | % (SE) | % (SE) | % (SE) |
|---|---|---|---|---|---|---|---|
| *Missing information in all visits* | | | | | | | |
| n | 6061 | 15774 | 57504 | 13422 | 4925 | 16895 | 114581 |
| Personal history (complete) | 99.3 (0.1) | 92.6 (0.2) | 67.9 (0.2) | 42.5 (0.4) | 62.1 (0.7) | 95.2 (0.2) | 73.8 (0.1) |
| Family history (complete) | 99.4 (0.1) | 92.6 (0.2) | 65.1 (0.2) | 39.7 (0.4) | 61.6 (0.7) | 95.9 (0.2) | 72.1 (0.1) |
| Personal history (any) | 88.2 (0.4) | 88.5 (0.3) | 56.4 (0.2) | 17.2 (0.3) | 54.2 (0.7) | 91.0 (0.2) | 62.9 (0.1) |
| Family history (any) | 97.5 (0.2) | 88.9 (0.3) | 57.3 (0.2) | 28.9 (0.4) | 53.5 (0.7) | 93.8 (0.2) | 65.7 (0.1) |
| Height | 42.7 (0.6) | 19.0 (0.2) | 15.6 (0.2) | 20.7 (0.3) | 30.6 (0.7) | 23.3 (0.3) | 19.9 (0.1) |
| Weight measurement | 35.8 (0.6) | 16.0 (0.2) | 18.8 (0.2) | 5.6 (0.2) | 8.9 (0.4) | 12.3 (0.3) | 16.4 (0.1) |
| Waist circumference | 49.5 (0.6) | 36.0 (0.3) | 15.6 (0.2) | 12.7 (0.2) | 36.9 (0.6) | 63.6 (0.4) | 27.8 (0.1) |
| Blood pressure | 1.8 (0.2) | 5.8 (0.2) | 9.8 (0.1) | 4.6 (0.2) | 4.2 (0.3) | 6.3 (0.2) | 7.4 (0.1) |
| Blood cholesterol | 87.0 (0.4) | 80.1 (0.3) | 81.9 (0.2) | 97.4 (0.1) | 91.1 (0.4) | 82.0 (0.3) | 84.2 (0.1) |
| Incomplete information | 99.6 (0.1) | 95.1 (0.2) | 98.1 (0.1) | 99.6 (0.1) | 99.6 (0.1) | 99.6 (0.1) | 98.2 (0.1) |
| *Missing information in first visits* | | | | | | | |
| n | 5103 | 10999 | 23053 | 4983 | 2298 | 6688 | 54224 |
| Personal history (complete) | 99.2 (0.1) | 89.9 (0.3) | 72.8 (0.3) | 35.1 (0.7) | 49.3 (0.9) | 92.6 (0.3) | 76.3 (0.2) |
| Family history (complete) | 99.3 (0.1) | 89.8 (0.3) | 71.3 (0.3) | 41.4 (0.7) | 50.9 (0.9) | 93.5 (0.3) | 76.4 (0.2) |
| Personal history (any) | 88.2 (0.5) | 85.1 (0.3) | 58.3 (0.3) | 15.3 (0.5) | 40.1 (0.8) | 86.9 (0.4) | 68.3 (0.2) |
| Family history (any) | 97.3 (0.2) | 85.2 (0.3) | 60.6 (0.3) | 30.0 (0.6) | 40.1 (0.8) | 90.5 (0.4) | 65.0 (0.2) |
| Height | 41.8 (0.7) | 19.0 (0.4) | 14.8 (0.2) | 21.1 (0.6) | 20.4 (0.7) | 23.0 (0.5) | 20.1 (0.2) |
| Weight | 35.3 (0.3) | 15.0 (0.3) | 16.7 (0.2) | 47.6 (0.3) | 7.2 (0.4) | 12.9 (0.4) | 15.9 (0.1) |
| Waist circumference | 48.9 (0.7) | 33.6 (0.4) | 11.1 (0.2) | 12.9 (0.5) | 23.0 (0.7) | 66.1 (0.6) | 26.9 (0.2) |
| Blood pressure | 1.7 (0.2) | 5.4 (0.2) | 4.9 (0.1) | 5.0 (0.3) | 3.2 (0.3) | 7.0 (0.3) | 4.9 (0.1) |
| Blood cholesterol | 86.1 (0.5) | 75.7 (0.4) | 76.6 (0.3) | 97.9 (0.2) | 92.3 (0.4) | 79.1 (0.5) | 80.6 (0.2) |
| Incomplete information | 99.6 (0.1) | 93.0 (0.2) | 96.7 (0.1) | 99.7 (0.1) | 99.4 (0.1) | 99.4 (0.1) | 97.01 (0.1) |
| *Risk factors screening in all visits* | | | | | | | |
| n | 3423 | 12015 | 45108 | 10484 | 3374 | 12750 | 87154 |
| BMI | | | | | | | |
| Normal | 48.3 (0.9) | 48.2 (0.5) | 51.3 (0.2) | 44.6 (0.5) | 52.3 (0.9) | 46.4 (0.4) | 49.3 (0.2) |
| Underweight | 4.9 (0.4) | 4.5 (0.2) | 8.0 (0.1) | 4.5 (0.2) | 9.1 (0.5) | 3.8 (0.2) | 6.3 (0.1) |
| Overweight | 31.9 (0.8) | 34.4 (0.4) | 30.8 (0.2) | 34.7 (0.4) | 29.1 (0.8) | 34.2 (0.4) | 32.3 (0.2) |
| Obese | 14.9 (0.6) | 12.9 (0.3) | 10.0 (0.3) | 16.3 (0.4) | 9.4 (0.5) | 15.7 (0.3) | 12.1 (0.1) |
| n | 5942 | 14835 | 51784 | 12773 | 4717 | 15814 | 105865 |
| Hypertension | 35.4 (0.6) | 28.0 (0.4) | 42.5 (0.2) | 33.7 (0.4) | 25.6 (0.6) | 35.9 (0.4) | 37.2 (0.1) |
| *Risk factors screening in first visits* | | | | | | | |
| n | 2925 | 8440 | 18820 | 3850 | 2678 | 5078 | 41791 |
| BMI | | | | | | | |
| Normal | 49.3 (0.9) | 48.0 (0.5) | 51.5 (0.4) | 45.3 (0.8) | 52.4 (1.0) | 44.4 (0.7) | 49.3 (0.2) |
| Underweight | 4.8 (0.4) | 4.5 (0.2) | 7.7 (0.2) | 5.1 (0.4) | 10.1 (0.6) | 4.2 (0.3) | 6.3 (0.1) |
| Overweight | 32.0 (0.9) | 34.2 (0.5) | 30.4 (0.3) | 33.3 (0.8) | 28.7 (0.9) | 35.1 (0.7) | 32.0 (0.2) |
| Obese | 13.9 (0.6) | 13.3 (0.4) | 10.4 (0.2) | 16.3 (0.6) | 8.8 (0.5) | 16.3 (0.5) | 12.4 (0.2) |
| n | 5008 | 10379 | 21858 | 4725 | 3288 | 6201 | 51459 |
| Hypertension | 34.5 (0.7) | 28.5 (0.4) | 40.5 (0.3) | 31.6 (0.7) | 25.1 (0.8) | 37.9 (0.6) | 35.3 (0.2) |

Within province, rural–urban comparisons are significant at 0.05.
Between provinces, comparisons are significant at 0.05.
Differences in proportion tested using $\chi^2$.
BMI, body mass index.

more common in urban districts of Central Java (table 2). However, these data should be interpreted cautiously due to the relatively high missing data on the measurements.

## Barriers for the screening of hypertension in POSBINDU

The qualitative data supported the quantitative finding about lacking participation of male and younger population in POSBINDU. In the FGDs, cadres and health officials stated the barriers for male and younger participants to attend POSBINDU, including the inconvenience of POSBINDU schedule, as well as low awareness for hypertension screening (table 3).

The need for role model from community leader to improve participation and the barrier for participation, particularly among men is highlighted by these quotes:

Yes, we don't have a lot of men (participants), because they are working (Cadre, FGD#21)

In our POSBINDU, the awareness for early screening is still low. Only several people come (to POSBINDU), younger people don't want to come because (POSBINDU is conducted) during working days (Cadre, FGD#3)

…Socialization for this (POSBINDU) is needed, often, the community leader in our area don't want to participate because they are afraid to be screened (Cadre, FGD#2)

… when I asked the communities, why they did not come to POSBINDU, or why there were only few people, they said because I (the community member) were not sick, so why do I need to get (health) check-up (?). So, they were not aware that POSBINDU is not only for those who are sick (Health official, FGD#19)

I asked POSBINDU (participant), why elderly? Where are the younger population? And they said that the young stayed at home because they were embarrassed if they have diseases… (Health official, FGD#16)

Lack of priority and overlap of NCD-related programmes also contribute to a suboptimal target population of POSBINDU, as illustrated by the following quote:

(NCD) is not a priority program, hence, there's a lack of commitment between the superior (health department) with the program officials, for example. (Health official, FGD#7)

… (different department in) the Ministry of Health focus on specific diseases, such as diabetes and cancer… However, in the community, (the programs) become general. (We) run Polindes, (Posyandu) Lansia, POSBINDU, School Health Program (Types of community based public health programs in Indonesia). In my opinion, the regulation is rigid and detailed, but the implementation is mixed (overlap). If we want to give optimum results, it takes efforts. (Health official, FGD#10).

Several barriers for implementation were revealed. The cadres and health officers often have to run

several different programmes. The FGDs also revealed a lack of capability of cadres to conduct measurements for hypertension screening, providing health education and also conducting the recording and reporting of the POSBINDU activities and measurements. The informants also mentioned a lack of resources, including budget, equipment and logistics to conduct all the measurements.

One person can hold 5 positions in PHC activities… POSBINDU cadres, Posyandu Lansia cadres, and other programs. (Cadre, FGD#11)

(cadres of) POSBINDU do not have laptop nor cell phone for the reporting application (of POSBINDU), hence, we report to PHC manually (Cadre, FGD#14)

The barriers also include the complexities of the activities and measurements, as well as extensive reporting forms, which require a long time to be completed.

…it takes a long time, because of the measurements and stages (of POSBINDU activities) (PHC officer, FGD#8)

…POSBINDU report is too time-consuming, because it is long (detail), including identity, cell phone number, address, and others… and it has to be filled out every month. (PHC officer, FGD#6)

Interestingly, in several districts, we found the implementation of mobile POSBINDU, moving from one community to the other within the same subdistricts.

Our POSBINDU is mobile, we have ten communities, so every week, we move from one community to the next, focusing on people 15–59 years old. (Cadre, FGD#18)

We further synthesised the quantitative and qualitative results. We categorised the barriers into three main parts: (1) input, reflecting the target population/coverage of POSBINDU, (2) process, describing the implementation of POSBINDU activities and (3) output, reflecting the recording and reporting process of POSBINDU (figure 2). Results show that in both approaches we found lacking participation of male and younger people in POSBINDU. Lack of priority for NCD screening and ineffective coordination among stakeholders, combined with lack of awareness and access might attribute to this finding. The high missed opportunity, particularly in history taking and measurements, were likely due to the complexity of the activities/measurements, as well as lack of resources. The high missing data also stem from the complexity of the forms and lack of capability for online reporting.

## DISCUSSION

In this study, we revealed missed opportunities in input, activities and output of POSBINDU implementation in screening for hypertension and its risk factors. Several contextual barriers were identified. The suboptimal coverage was possibly due to lack of priority for NCD screening, lack of awareness and access and overlap of NCD-related programme. The

**Table 3** Qualitative analyses of focus group discussion among POSBINDU cadres, primary health care and health department officials

| Themes | Category | Codes |
|---|---|---|
| Suboptimal target population and gap in policy | Participants' characteristics | Younger adults rarely participate |
| | | Lack of male participants |
| | Barrier to participations | Schedule incompatibility |
| | | Low awareness for screening |
| | | Lack of role model for screening |
| | Ineffective policy and coordination | Lack of prioritisation for NCD |
| | | Implementation gap of national policy/programme at the local level |
| | | The need for coordination with different stakeholders |
| | | The need for coordination among NCD-related programmes |
| Lack of human resources in terms of capability and quantity for hypertension screening | Cadres have multiple tasks, with time constraints | POSBINDU cadres often have to multitask and handling other community programmes |
| | | Cadres are volunteers with other obligations |
| | Cadres' competencies | Lack of knowledge on hypertension and other NCD |
| | | Lack of ability to conduct measurements and provide health education |
| | | Lack of ability to conduct recording and reporting |
| | Lack of NCD programme officers for supervision and reporting | Lack of NCD programme officers at PHC |
| | | Most programme officers are responsible for multiple tasks/ programmes |
| | | Lack of reporting officers |
| | Provision of referral counselling | The participant with hypertension is not always referred to PHC |
| | | Lack of counselling to participants before the referral made |
| | | POSBINDU has referral form, but rarely used |
| | | Treatment for the referral is covered by their health insurance |
| Lack of resources for hypertension screening and prevention | Equipment for hypertension screening | The equipment is sometimes incomplete |
| | | Equipment maintenance is inadequate |
| | | Limited logistics for cholesterol measurement |
| | Lack of budget | POSBINDU is funded by the government, stakeholder (private sectors) or community |
| | | Lack of budget for POSBINDU activities |
| | | Lack of budget for cadres training and incentives |
| | Health education material | Lack of health education materials |
| | Infrastructure for recording and reporting | Not all cadres have laptops |
| | | Limited internet connection in some areas |
| | | Most POSBINDU stations use manual reporting |
| Time constraints for implementation based on MOH standard | The complexity of activities and time limitation | The time required for examination is too long |
| | | Too many information needs to be asked and filled out |
| | | The referral form is rarely used |
| | The complexity of reporting forms | Many forms need to be filled, while time is limited |
| | | A simplified form in checklist format is preferred |

MOH, Ministry of Health; NCD, non-communicable disease; PHC, primary healthcare.

suboptimal activities and reporting were likely caused by a lack of resources, as well as limited time to perform the complexities of activities and reporting according to MOH guideline.

The missed opportunity to screen male and younger population that we found in this study is particularly concerning. Although the prevalence is lower than of the older population, hypertension prevalence among young Indonesian is

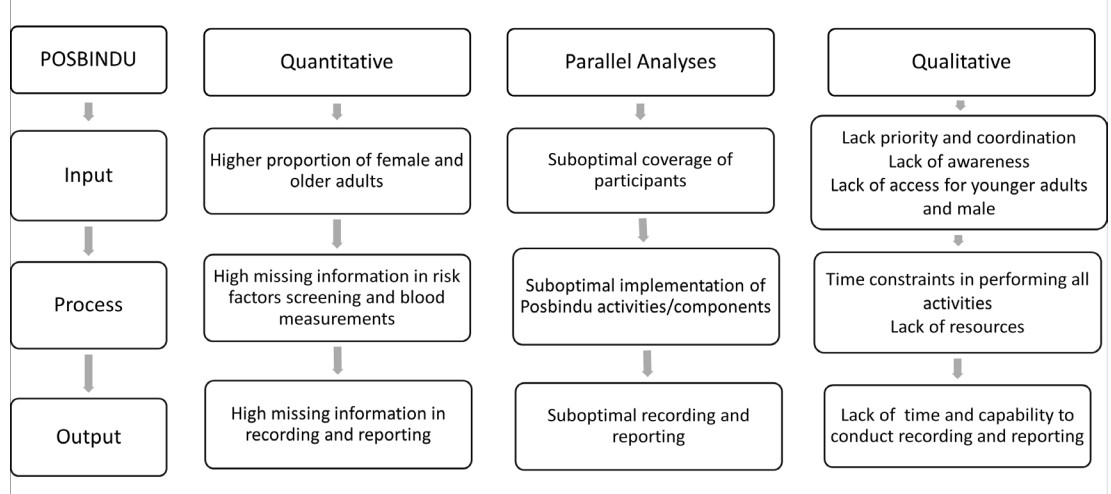

**Figure 2** Synthesis of the quantitative and qualitative findings.

still relatively high (28%).[26] While the target population of POSBINDU is listed as those 15 years or older, the elderly are usually targeted in Posyandu Lansia, a community-based screening and management for the elderly population.[27] Awareness is also lower in male and younger adults, signalling the need to screen this population.[3] Ideally, POSBINDU becomes the 'gatekeeper' for screening in the community. Patients with hypertionsion and diabetes were then referred to PHC and joined Prolanis, a community-based activity funded by the health insurance programme, for management of chronic diseases patients.[12]

Furthermore, with a lack of male participation, POSBINDU is missing one of the key target populations for risk factors screening: smokers. Analysis of a national survey in 2014 reported 32% prevalence of smoking, with approximately 40% of males aged 15–55 years old and 14% of male adolescents are current smokers.[28–30] Further, 20% of Indonesia's total chronic diseases are attributed to smoking, with hypertension as the highest proportion.[31] Screening for hypertension and its risk factors earlier, combined with lifestyle-based interventions effectively avoid future complications.[32 33]

We also revealed the need to prioritise and reorganise the current NCD-related programmes, to address the suboptimal coverage and the overlap. An example of the gap between the national recommendation and local implementation is reflected in the coordination of existing NCD-related programmes: POSBINDU, Posyandu Lansia and PANDU PTM. In the MOH, the PANDU PTM (Pelayanan Terpadu Penyakit Tidak Menular, Integrated Health Services for NCDs) and POSBINDU are regulated under the Directorate for Disease Management, while Posyandu Lansia is under the Directorate of Public Health. Despite the different directorates, the implementation at community level is often conducted simultaneously and often overlap. Reporting, however, is conducted separately. Hence, as previous studies have noted, we also recommend the need of comprehensive and coordinated NCDs prevention programme in Indonesia.[34–36]

The relatively high missed opportunity in screening for hypertension risk factors, as well as sociodemographic characteristics found in this study, portrays suboptimal implementation of POSBINDU. This can be caused by a lack of recording and reporting (monitoring and evaluation fidelity) or lack of measurement (implementation fidelity). In our further elaboration during the FGDs, we found that lack of human resources might contribute to the suboptimal implementation of POSBINDU. Our findings revealed the need to train cadres to improve their skills and efficiency in conducting the measurements and history taking, as well as reporting the measurements. This is in line with findings from Meinema *et al* and Abdell-All *et al*.[37 38] Our findings also imply the complexities of the activities and reporting of POSBINDU which lead to ineffective implementation. It is important to ensure that valuable screening information can be recorded and followed up, for better intervention. A simplified screening programme with integrated reporting is needed.

In this study, we also discovered lack of financial resources and equipment as barriers to POSBINDU implementation. The integration of POSBINDU and PANDU PTM to the national health insurance scheme might be important to ensure the sustainability of funding for the programme. Integration of POSBINDU into the national health insurance can also improve participation of the working population, most of whom are covered by the national health insurance.[39] Previous studies have reported an increase in uptake of service by health insurance membership.[40–42]

Based on our findings, we identified two main areas that needs to be improved: coverage and implementation of POSBINDU. To improve coverage of POSBINDU, there are two important steps that we recommend. First, an integrated approach with collaboration among different programmes and directorates to reduce the overlap and simplify the POSBINDU implementation at the PHC and community level. PANDU PTM as the adaptation of WHO PEN,[43] needs to be implemented in a wider scale. Second, redirecting the

target population of hypertension screening, to cover also younger and male population. A workplace-based screening programme which can address the barriers identified in the qualitative findings is recommended.[44 45] For this younger population, the use of mobile technology for monitoring of risk factors and measurement might be effective. Previous studies have reported the effectiveness of mobile health for hypertension screening and risk stratification.[46 47]

To improve the implementation and components of POSBINDU activities, a simplified algorithm to screen and refer the target population is needed. The algorithm needs to be developed both in the electronic format and manual format to address the different capabilities of community cadres and resources in the community. Simplifying the programme and reporting systems will also reduce the workload of PHC and district health officials. Further, a clear algorithm for the referral of 'screened' cases to PHC is important. The readiness of the PHCs also needs to be improved to adequately manage the potential surge in referred cases. Lastly, there is a need to integrate hypertension and CVD screening programme into the national health insurance system. Hence, ensuring the sustainability of funding and resources of the programme. With these approaches, comprehensive screening for hypertension and CVD along the continuum of care might be more effective.

This study has several limitations. First, the proportion of our measures are not reflective for the whole target population of POSBINDU, since the participants were mostly female and of older age. The characteristics of our sample, which are generally older with a higher proportion of females, drive the proportion of risk factors higher than that of the general population in Indonesia. However, this study reflects the current participants of POSBINDU. Second, we used a secondary data collection by POSBINDU cadres, the high number of missing data that we presented in this study, probably stem from two main sources: omissions in reporting or a true lack in measurement/activities. Nevertheless, both the activities and reporting are important in NCDs screening, particularly in the follow-up. The secondary data also prone to measurement bias, particularly, with the variations in POSBINDU measurements by cadres. The MOH provided guidelines in the measurement for hypertension in POSBINDU, however, the implementation might vary. The high missing information on several sociodemographic characteristics that is, occupation and education, also limit our ability to conduct multivariable analyses. Another limitation of this study is we have not included the perspective of POSBINDU participants in the FGDs. Instead, we considered the POSBINDU cadres to represents the voice of both the implementers as well as users. However, we include the perspective of the POSBINDU participants as users in the baseline of our prospective data collection (ongoing). The users' perspective can provide further insights into barriers and facilitators of POSBINDU implementation.

Despite the limitation, there are several strengths of the study: First, to our knowledge, this was the first relatively large evaluation of POSBINDU. Second, the use of a mixed-methods study design, and therefore, providing more comprehensive information on POSBINDU implementation. Third, the study also investigates the contextual factors that should be addressed in the improvement of the community-based hypertension screening programme in Indonesia. This study might provide insights into POSBINDU implementation in other areas in Indonesia and can be the basis for further recommendation to improve POSBINDU implementation.

## CONCLUSION

This study showed the suboptimal implementation of POSBINDU activities. Particularly, the missed opportunity in screening for hypertension risk factors in Indonesia. The barriers include a lack priority for NCDs, lack of awareness and access for subpopulation, and several implementation barriers: capability, resources and protocols. An innovative approach to simplify and improve the capacity of POSBINDU is in preparation to optimise the screening and linkage to hypertension care in Indonesia. This study provides evidence-based recommendations in improving the current implementation of POSBINDU, in the Indonesian context.

**Acknowledgements** We thank SUNISEA project consortium and the district health departments, primary healthcare staffs and cadres of POSBINDU in the three provinces, who have contributed to this study.

**Collaborators** Scaling Up Non-Communicable Disease Intervention in South East Asia (SUNISEA) Project: Maarten Postma - University Medical Center Groningen, Robert Lensink - University of Groningen, Martin Rusnák - Trnavska Univerzita v Trnavepic. Caitlin Littleton - HelpAge International, Anil Krisna - HelpAge International, Michael Grimm - Universität Passau, Thi-Phuong-Lan Nguyen - University of Medicine and Pharmacy, Thai Nguyen University, Tran Thi Mai Oanh - Health Strategy and Policy Institute.

**Contributors** VW, AP, RPF, EPP, YAM, JK and JL-D contributed to the design of the study. VW, AP, RPF, EPP, YAM, BB and SS participate actively in study implementation, including in data collection. VW, YAM, SS and RPF analysed the quantitative data. AP, EPP and BB analysed the qualitative data. VW, AP, RPF, SS and YAM draft the manuscript with all co-authors revised critically. All authors read and approved the final manuscript. The corresponding author is respobsible for the overall content as guarantor.

**Funding** The SUNI-SEA research project was funded by the European Union's Horizon 2020 research and innovation programme, call SC1-BHC-16-2018 Global Alliance for Chronic Diseases (GACD) - Scaling-up of evidence-based health interventions at population level for the prevention and management of hypertension and/or diabetes, soliciting for research in Low- and Middle-Income Countries (LMIC), under grant agreement No:825026. The funding source was not involved in the data collection, data analysis, manuscript writing and publication.

**Competing interests** None declared.

**Patient and public involvement** Patients and/or the public were not involved in the design, or conduct, or reporting, or dissemination plans of this research.

**Patient consent for publication** Not applicable.

**Ethics approval** This study involves human participants and was approved by the ethical review board at Universitas Gadjah Mada, reference number KE/FK/0648/2019. Participants of focus group discussion gave informed consent to participate in the study before taking part.

**Provenance and peer review** Not commissioned; externally peer reviewed.

**Data availability statement** Data are available upon reasonable request. The de-identified data from this study are available upon request to the corresponding or first author pending thorough review of request and adherence to the Indonesian government regulation on data sharing.

**ORCID iDs**
Vitri Widyaningsih http://orcid.org/0000-0003-0116-7120
Ari Probandari http://orcid.org/0000-0003-3171-5271

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
