## [Reviewer comments · BMJ Open]

ARTICLE DETAILS

TITLE (PROVISIONAL)	Missed Opportunities in Hypertension Risk Factors Screening in Indonesia: A Mixed-methods Evaluation of Integrated Health Post (POSBINDU) Implementation
AUTHORS	Widyaningsih, Vitri; Febrinasari, Ratih; Pamungkasari, Eti; Mashuri, Yusuf; Sumardiyono, Sumardiyono; Balgis, Balgis; Koot, Jaap; Landsman-Dijkstra, Jeanet; Probandari, Ari

VERSION 1 – REVIEW

REVIEWER	Herman, Bumi Chulalongkorn University, Public Health
REVIEW RETURNED	04-Jun-2021

GENERAL COMMENTS	Thank you very much for the opportunity to review this article, This article has an objective to assess the implementation and contextual barriers of POSBINDU in Indonesia however, it seems that the article elaborates on hypertension screening only, therefore please re-write the objective. The quantitative part identified the discrepancy of visits between gender, and the underlying reason was explained in the Qualitative part. However, some issues need to be clarified, including the clear term of outcome and detailed information on how the qualitative part was conducted. The author can express their opinion or even rebuttal suggested comments. Full comments are summarized in a file attached with the review. Methodology: Quantitative section 1. What are the concise the definition of rural and urban area in this study, and how the researchers assure that the character of rural areas in the densely-populated area was similar with the rural areas in the less-populated region?2. As for the definition of the outcome, the component of hypertension screening consists of history taking, anthropometric measurement, blood pressure, and cholesterol. It seems that in this study, the author analyzed each element of screening and showed the proportion of each element. Why don't the authors set a binary outcome where the definition of complete screening is receiving those 4 services? Also, If one participant missed one of these four components, was it consider missed screening? In practice, a person can be diagnosed with hypertension if there is a consistent high systolic and/or diastolic blood in two different measurements on a visit. Hence, history of taking and blood pressure measurement are two essential components of screening. Was there any difference in missed screening proportion when applying two components of screening instead of four? Here is the example a. Person 1 attend 4 services: not missed the screening
---

b. person 1 missed the cholesterol test: missed the screening

3. It seems that in POSBINDU, there was heterogeneity in measuring blood pressure to diagnose hypertension, either using an automatic sphygmomanometer or using a manual one that relies on Korotkoff sound. Also, there was no information on who performed the measurement and how the situation of the POSBINDU during the measurement. Kindly state this as a limitation.

4. Could the author elaborate what is the definition of “complete personal history” and “any personal history”? this question also applied to the family history. And also what are the other information provided in the cohort (such as education, or occupation?)

5. As for statistical analysis, it is challenging to conclude the analysis and the footnote in Table 2 did not identify which factors that have the association. Was the author planned to have further tests including logistic regression or not? If not, please explain

Qualitative method

1. The author kindly fill the checklist for qualitative study (COREQ) and insert some essential information according to the checklist as a supplement, including three important domains to justify that the qualitative part was conducted properly. Please follow this checklist Allison Tong, Peter Sainsbury, Jonathan Craig, Consolidated criteria for reporting qualitative research (COREQ): a 32-item checklist for interviews and focus groups, International Journal for Quality in Health Care, Volume 19, Issue 6, December 2007, Pages 349–357, This checklist will answer several questions that are important but not concise enough to be included in the text including these issues:

a. Domain of Researchers, who and what were the backgrounds of interviewer and FGD moderator

b. How the participants recruited and why recruiting these type of participants

c. why the number of FGD participants is too small (4) or even too large (18) What were the justifications to determine the number of participants?

d. The methodology to ensure the validity of the statement. It seems that in this article, only the provider’s perspective was seen. For example, the low proportion of people who underwent screening in males is due to inconvenient time and POSBINDU was conducted during working hours. It needs a triangulation technique to confirm this finding, which we should clarify from the client’s side of why men did not attend the screening. Perhaps we could find another answer such as they might feel uncomfortable with the PUSBINDU system or even have a personal issue with the PUSBINDU staff.

Another statement from cadre 2 on FGD stated “ ... Socialization for this (hypertension screening) is needed, often, the community leader in our area don’t want to participate because they are afraid to be screened” (Cadre, FGD#2) This needs a triangulation to test the validity of this statement. The health cadre as the provider’s perspective told us about this, however, we have no idea of why the client is afraid to get screened.

There are some ways to do triangulation about one aspect, One FGD is conducted for POSBINDU providers and the other FGD is made for POSBINDU clients. One similar topic is discussed in both groups then synthesized to yield a valid statement. If the researcher did this already, kindly revised the article accordingly and include the cited statement.

2. what is the basic theory to derived the FGD guideline/theme to assess the reason for missed screening for example health service delivery model in primary care? Please state in the text so the readers will have an insight into how the theme was derived

	3. The author should state in the methodology that this study conducted a mixed-method study using a sequential explanatory method. Issues in statistics 1. In this article, it is also important to state the reason to discretize or classify the age and number of visits. There is a different range in age per subset variables such as 15-24 (10 unit) and 25-40 (15 unit). Please justify the reason for performing this classification. This could also apply to the number of visits. Otherwise, the author could use the mean instead. 2. If possible, could the author elaborate on the bivariate results of these variables in Stata output such as each test and p-value? a. Association of rural and urban area versus missed hypertension screening b. age and missed screening c. number of PHC and missed hypertension screening d. number of POSBINDU and missed hypertension screening e. Gender and missed hypertension screening and whether these variables will be tested for logistic regression to find the associated factors of missed hypertension screening. If not please explain
--	---

REVIEWER	Boateng, Daniel UMC Utrecht
REVIEW RETURNED	22-Aug-2021

GENERAL COMMENTS	This is an important study, assessing the implementation and contextual barriers of the Integrated Health Post (POSBINDU) in Indonesia. Generally, the article is well written. Below are comments to improve it. Main comments  - Kindly add a flowchart to explain the participant recruitment process. - The methods omit an important section on how the key variables were measured. - What kind of statistical differences did you explore using Chi-square, T-test, and ANOVA? - Did you employ any theoretical approach in integrating the findings from the qualitative and quantitative? - What is SE in tables 1 and 2? It was not mentioned anywhere in the methods or results and not defined under the table as a footnote. Other abbreviations should all be defined under the table. - What was the basis for the classification of the number of visits in Table 1? Minor comments Define POSBINDU in the abstract Non-Communicable should be "Non-communicable" Page 2, line 22: These "figures" should be "This figure" Page 2, line 26; Hypertension "is accounted for" should be "accounts for" "low-middle income countries": were you referring to "low- and middle-income countries?"
--

REVIEWER	Rahmawati, Riana
-----------------	------------------

	Islamic University of Indonesia
REVIEW RETURNED	06-Sep-2021

GENERAL COMMENTS	Thanks for the opportunity to review this manuscript. Generally, this manuscript is well written, well-structured and interesting to read. However, I have some concerns regarding the data presentation as detailed in the following:  - Title: What is the English term for POSBINDU? In the title, the authors used “Integrated Screening Post,” but on Page 2 (line 33) they used Integrated Health Post. - In the Methods (Setting), it is said, “POSBINDU aims to empower communities in screening for NCDs and the risk factors, targeting individuals above 15 years old, particularly those of productive age.[19,20] The main activities include screening for NCDs (mainly hypertension and diabetes) and the risk factors (i.e., smoking, diet, physical activity, obesity)”. These statements reflect the role of POSBINDU in the screening of NCDs’ risk factors, including hypertension. However, the authors highlighted “missed opportunity in hypertension screening and risk factors” throughout the paper. What the “risk factors” refer to? Are they CVD or hypertension risk factors? We know that some CVD risk factors are also factors affecting hypertension. The authors, therefore, should carefully present and highlight the main findings. - Data collection. There were 2 in-depth interviews. Does it mean 2 participants being interviewed? Who are they? - Outcome (line 36-37). According to the study objective, the primary outcome of this study is POSBINDU implementation instead of missed opportunities in hypertension screening - Table 1 shows that 23,053 out of 54,224 (around 60%) of participants came from urban North Sumatera. The number of Posbindu in this region is the highest among other cities. Please briefly explain in the Methods section. - This study revealed a high proportion of missing information on personal and family history. Please briefly explain the kinds of information gathered for these variables. - Table 2. (Risk factor screening). I assume the data report the NCD risk factors instead of hypertension risk factors. Why did the authors include two factors only (obesity and hypertension). What about diabetes and high cholesterol? - Page 7, line 43. socialization for this (hypertension screening)..... (comment: hypertension or NCD screening?) - Page 5 line 57: this paragraph explains the relatively high missing information for hypertension screening. On page 6 (line 3), however, it is said, “we found the highest proportion of available data for blood pressure measurements in all the seven districts. These are contradictory statements. It would be fine if the first sentence changed into “..high missing information for NCD risk factors’ screening”. As mentioned in the Abstract, this study aimed to assess the implementation and contextual barriers of POSBINDU in Indonesia. The first sentence of the Discussion also says “In this study, we revealed missed opportunities in input, activities, and output of POSBINDU Implementation” (page 8 line 46). In the
--

	Discussion, the authors have explained well the contextual barriers related to the NCD screening. With regard to hypertension screening, the authors need to highlight the finding that showed almost all visitors had had blood pressure examination. The statement “the relatively high missed opportunity in hypertension screening...” (page 9, line 27) therefore need to be evaluated. - Page 9, line 40. What does PANDU PTM stand for? - Conclusion. Again, the statement regarding the missed opportunities for hypertension screening needs to be reworded.
--	---

VERSION 1 – AUTHOR RESPONSE

Reviewer: 1 Dr. Bumi Herman, Chulalongkorn University Comments to the Author: Thank you very much for the opportunity to review this article, This article has an objective to assess the implementation and contextual barriers of POSBINDU in Indonesia however, it seems that the article elaborates on hypertension screening only, therefore please re-write the objective. The quantitative part identified the discrepancy of visits between gender, and the underlying reason was explained in the Qualitative part. However, some issues need to be clarified, including the clear term of outcome and detailed information on how the qualitative part was conducted. The author can express their opinion or even rebuttal suggested comments. Full comments are summarized in a file attached with the review.	Thank you for your valuable feedback, which help us in improving the quality of this paper.
Methodology: Quantitative section 1. What are the concise the definition of rural and urban area in this study, and how the researchers assure that the character of rural areas in the densely-populated area was similar with the rural areas in the less-populated region?	Thank you for your detailed insight. The rural-urban areas definition followed the Indonesian National Bureau of Statistics regulation Number 120, year 2020, which is based on several indicators: number of population/km2, proportion of families who has main occupation in agriculture, proportion of families with access to internet/phone and electricity, and availability of public facilities: school, traditional market, hospital, and entertainment. We add part of this information in the text for further clarification. Line 103-104 “The rural/urban classification is based on population density and facilities available in the communities.”
2. As for the definition of the outcome, the component of hypertension screening consists of history taking, anthropometric measurement, blood pressure, and cholesterol. It seems that in this study, the	Thank you for your suggestion. We analysed each outcome individually to

author analyzed each element of screening and showed the proportion of each element. Why don't the authors set a binary outcome where the definition of complete screening is receiving those 4 services? Also, If one participant missed one of these four components, was it consider missed screening? In practice, a person can be diagnosed with hypertension if there is a consistent high systolic and/or diastolic blood in two different measurements on a visit. Hence, history of taking and blood pressure measurement are two essential components of screening. Was there any difference in missed screening proportion when applying two components of screening instead of four? Here is the example a. Person 1 attend 4 services: not missed the screening b. person 1 missed the cholesterol test: missed the screening	provide more detailed information to the health department, as well as for further improvement of the program. We add the additional rationale on the methods section. Line 122-124) "Analyses was conducted on each indicator to provide more detailed information on specific components of screening which was lacking." We valued your feedback, and think that this is an important evaluation on POSBINDU implementation. Hence, we also generate a new variable (incomplete information) which represent whether the individual received the recommended procedure (history taking, anthropometric measurement, blood pressure measurement, and blood examination). We add this information in both the methods section and results (table and narrative). Line 133-136) "We also generate variable "incomplete information" which represent whether the individual received the recommended procedure (history taking, anthropometric measurement, blood pressure measurement, and blood examination). The proportion presented in the analyses described the individuals who did not receive the complete recommended procedure."
3. It seems that in POSBINDU, there was heterogeneity in measuring blood pressure to diagnose hypertension, either using an automatic sphygmomanometer or using a manual one that relies on Korrotkoff sound. Also, there was no information on who performed the measurement and how the situation of the POSBINDU during the measurement. Kindly state this as a limitation.	Thank you, we add the information as an additional limitation. Since the study was based on secondary data of POSBINDU records, data on how measurement was conducted was lacking. However, we add information on the SOPs of hypertension screening in POSBINDU as stated by the Ministry of Health in the

	POSBINDU guidelines. (Line 357-360) “The secondary data also prone to measurement bias, particularly, with the variations in POSBINDU measurements by cadres. The Ministry of Health provided guidelines in the measurement for hypertension in POSBINDU, however, the implementation might vary.”
4. Could the author elaborate what is the definition of “complete personal history” and “any personal history”? this question also applied to the family history. And also what are the other information provided in the cohort (such as education, or occupation?)	Thank you for your detailed feedback. We add information on variables measurement, including complete and any personal history in the methods section. (Line 128-132) “Personal and family history of NCDs were also obtained, which include seven (7) diseases: hypertension, diabetes, heart disease, stroke, asthma, cancer, and high blood cholesterol. Complete personal/family history variables were coded 1 if all information was available and coded 0 if at least one of the disease histories was missing. Any personal/family history variables were coded 1 if at least one of the disease histories was available and coded 0 if all of the history information was missing.” We add information on level of education, although it has high missing value (59%). However, due to the reporting of the secondary dataset, data on occupation had high missing value (more than 60%). Hence, we did not analyse the variable on the reported manuscript. We add this information on the methods section, “Outcome and variables measurements”, and adding information on occupation and education:

	(Line 123-124) “Occupation was not included in the analyses due to high missing value in the POSBINDU report (>60%).”
5. As for statistical analysis, it is challenging to conclude the analysis and the footnote in Table 2 did not identify which factors that have the association. Was the author planned to have further tests including logistic regression or not? If not, please explain	This paper aimed to describe the process evaluation of POSBINDU implementation. With the use of secondary data, which has relatively high missing value, we were unable to ascertain factors related to the missing information quantitatively. Hence, logistic regression was not conducted. Instead, we provide potential explanation from the FGDs by cadres and community health workers. We add this also in the limitation section. (Line 360-362) “The high missing information on several sociodemographic characteristics i.e., occupation and education, also limit our ability to conduct multivariable analyses.”
Qualitative method 1. The author kindly fill the checklist for qualitative study (COREQ) and insert some essential information according to the checklist as a supplement, including three important domains to justify that the qualitative part was conducted properly. Please follow this checklist Allison Tong, Peter Sainsbury, Jonathan Craig, Consolidated criteria for reporting qualitative research (COREQ): a 32-item checklist for interviews and focus groups, International Journal for Quality in Health Care, Volume 19, Issue 6, December 2007, Pages 349–357, This checklist will answer several questions that are important but not concise enough to be included in the text including these issues: a. Domain of Researchers, who and what were the backgrounds of interviewer and FGD moderator	Thank you for your feedback. We add additional information regarding the qualitative data collection. Among the researchers, VW, EPP, JL and AP were public health researchers experienced in qualitative research. AP has been focusing on health systems research. EPP mainly an expert for health professional education. The FGD facilitators include VW, EPP, AP, RFP, S, and B. Two additional FGD facilitators were also recruited, with public health background and experience in conducting qualitative research. All facilitators attend the preparatory meeting to discuss the FGDs and interview guidelines, to obtain similar perception regarding the aims of FGDs and interviews and items of the

	FGD guidelines. (Line 116-117) We also add part of this information in the text “The FGD facilitators had public health background and experience in conducting qualitative research. All facilitators attend the preparatory meeting to discuss the FGDs and interview guidelines, to obtain similar perception regarding the aims of FGDs and interviews and items of the FGD guidelines.”
b. How the participants recruited and why recruiting these type of participants	Participants of FGDs includes health officials from health department and primary healthcare. Within each district that were included in this study, we invite health officials responsible for POSBINDU program from the district’s health department, and primary health care. We also invite 2-3 cadres from each PHC based on list of cadres obtained from PHC officials. These participants were recruited to obtain information on Posbindu implementation facilitators and barriers from the health systems supply sides. We also include community health volunteers or cadres who run and also participate in Posbindu. These participants were recruited to obtain information and perspective not only from the supply sides, but also demand sides. Since Posbindu cadres can provide insights into the participants perception, as they are usually also participate in Posbindu. We add this information on methods section Line 116-121 “The two in-depth interviews

	were conducted with health districts department officials. Within each district, we conducted purposive sampling to recruit health officials responsible for POSBINDU program from the district's health department, and primary health care. We also recruit 2-3 cadres from each PHC based on list of cadres obtained from PHC officials. These participants were recruited to obtain information on POSBINDU implementation facilitators and barriers. "
c. why the number of FGD participants is too small (4) or even too large (18) What were the justifications to determine the number of participants?	In our protocol, we aims to have approximately 6-12 people within each FGD. However, during two of the FGDs, there were more participants came to the venue and keen to participate in the discussions. Hence, there were two FGDs that had 17 and 18 participants, respectively. Both FGDs were among community health workers or cadres. Meanwhile the FGD with four participants happened when two of invited health officials were unable to attend FGDs due to sudden scheduling conflict. Hence, we proceed with 4 participants, and interviewed the two health officials another time individually. The other FGDs run as planned.
d. The methodology to ensure the validity of the statement. It seems that in this article, only the provider's perspective was seen. For example, the low proportion of people who underwent screening in males is due to inconvenient time and POSBINDU was conducted during working hours. It needs a triangulation technique to confirm this finding, which we should clarify from the client's side of why men did not attend the screening. Perhaps we could find another answer such as they might feel uncomfortable with the PUSBINDU system or even have a personal issue with the PUSBINDU staff.	Thank you for your feedback and suggestion. We add additional quotes supporting the sentence, as well as provide additional reference from previous publication. For the FGDs, we recruited health provider (districts health departments as well as primary health care provider). Meanwhile, cadres/community health volunteer represents both the provider and receiver (clients), since usually these cadres were also

	participants of POSBINDU. Additional quotes were also added for triangulation. We also conducted member checking, and add this information on methods section: Line 160-163 “ To enhance trustworthiness, we assess barriers of POSBINDU from the several sources for triangulation purposes: health and PHC officials to reflect implementer’s perspective, and cadres to reflect implementers and users’ perspective. During data analyses, we also discuss the findings with representative of the FGD participants, i.e., member checking.”
Another statement from cadre 2 on FGD stated “ ...Socialization for this (hypertension screening) is needed, often, the community leader in our area don’t want to participate because they are afraid to be screened” (Cadre, FGD#2) This needs a triangulation to test the validity of this statement. The health cadre as the provider’s perspective told us about this, however, we have no idea of why the client is afraid to get screened. There are some ways to do triangulation about one aspect, One FGD is conducted for POSBINDU providers and the other FGD is made for POSBINDU clients. One similar topic is discussed in both groups then synthesized to yield a valid statement. If the researcher did this already, kindly revised the article accordingly and include the cited statement.	Thank you for your feedback. We would like to clarify that we also consider that health cadres perspectives represent their views as an implementer of the POSBINDU, but also as the users. Additionally, there were information obtained from cadres or PHC officials about reasons of communities for not attending POSBINDU that they received from participants, prior to the FGDs. However, we realized that we need to involve the voices from community who do not visit POSBINDU. We have not included that in the phase of study, however we included that in the baseline phase before the interventions. Hence, we add some text to clarify this issue in the discussion. Line (362-365) “ Another limitation of this study is we have not included the perspective of POSBINDU participants in the FGDs. Instead, we considered the POSBINDU cadres to represents the voice of both the

	implementers as well as users. However, we include the perspective of the POSBINDU participants in the baseline of our prospective data collection (ongoing).” For triangulation, we synthesise the voices from several Cadres and PHC staff, including those sharing the perspective of communities based on their previous interaction with the communities for example on Line 232-238) “.. when I asked the communities, why they did not come to POSBINDU, or why there were only few people, they said because I (the community member) were not sick, so why do I need to get (health) check-up (?). So, they were not aware that POSBINDU is not only for those who are sick” (Health official, FGD#19) “ I asked POSBINDU (participant), why elderly? Where are the younger population? And they said that the young stayed at home because they were embarrassed if they have diseases.. “ (Health official, FGD#16)
2. what is the basic theory to derived the FGD guideline/theme to assess the reason for missed screening for example health service delivery model in primary care? Please state in the text so the readers will have an insight into how the theme was derived	Thank you for your feedback. We used the logic model framework for process evaluation to assess the implementation of POSBINDU. We adopt several indicators from the current literature on the use of logic model in process evaluation of community-based health intervention (Sharma et al., 2017; Smith et al., 2020; Wong et al., 2010). The FGDs theme as well as indicators of the secondary data that we developed based on the literature,

	were discussed with officials from health department and PHC officials in one pilot site for finalization. (Line 137-141) “We used the logic model framework for process evaluation to assess the implementation of POSBINDU. We adopt several indicators from the current literature on the use of logic model in process evaluation of community-based health intervention [22–24]. The FGDs theme as well as indicators of the secondary data developed based on the literature, were discussed with officials from health department and PHC officials in one pilot site for finalization.”
3. The author should state in the methodology that this study conducted a mixed-method study using a sequential explanatory method.	Thank you for the feedback. This was a concurrent mixed-methods study, in which we conducted the quantitative and qualitative data collection parallely. (Line 94) “This was a concurrent mixed-methods study in seven districts in three provinces in Indonesia”
Issues in statistics 1. In this article, it is also important to state the reason to discretize or classify the age and number of visits. There is a different range in age per subset variables such as 15-24 (10 unit) and 25-40 (15 unit). Please justify the reason for performing this classification. This could also apply to the number of visits. Otherwise, the author could use the mean instead.	Thank you for your feedback, We reclassified age to better represents the target population of Posbindu (15-59 years old) and classification by the Indonesian ministry of health. (Line 124-127) “Age was classified into several groups based the Indonesian Ministry of Health classification for age (youth = 15-24 years old, adult = 25-44 years old, pre-elderly = 45-59 years old, and elderly => 60 years old).”
2. If possible, could the author elaborate on the bivariate results of these variables in Stata output such as each test and p-value? a. Association of rural and urban area versus missed hypertension screening	Thank you for your suggestion and feedback. We added information on whether the differences

b. age and missed screening c. number of PHC and missed hypertension screening d. number of POSBINDU and missed hypertension screening e. Gender and missed hypertension screening and whether these variables will be tested for logistic regression to find the associated factors of missed hypertension screening. If not please explain	between rural and urban were significant at p 0.05 by adding information on notes after Table 1. However, we did not do logistic regression with the available data, due to relatively high missing value, particularly for age (> 22%). Additionally, the additional information and analyses might be too dense to be included in this manuscript, and beyond the scope of this paper. Hence, for this paper we focus on describing the missed screening for risk factors of hypertension.
Reviewer: 2 Dr. Daniel Boateng, UMC Utrecht Comments to the Author: This is an important study, assessing the implementation and contextual barriers of the Integrated Health Post (POSBINDU) in Indonesia.	Thank you for your feedback and comment. We truly appreciate your detail insight to our paper.
Generally, the article is well written. Below are comments to improve it. Main comments - Kindly add a flowchart to explain the participant recruitment process.	Thank you, we add flowchart of data collection to further clarify the methods (Figure 1)
- The methods omit an important section on how the key variables were measured.	Thank you, we added this information on the methods section
- What kind of statistical differences did you explore using Chi-square, T-test, and ANOVA?	Yes, we used Chi-square to test differences in proportion, and T-Test or ANOVA to test differences in mean. To make the information clearer, we add the specific test we use, on the table notes.
- Did you employ any theoretical approach in integrating the findings from the qualitative and quantitative?	We did not use specific theory to integrate the finding. However, we use the weaving technique, in which we were writing the findings of qualitative and quantitative together by concept or theme (Fetters et al., 2013) We add this information in the methods section (Line 152-153) “ Weaving technique, analyzing the quantitative and qualitative findings

	together by theme or concept, was used to integrate the findings.(Fetters et al., 2013)”
- What is SE in tables 1 and 2? It was not mentioned anywhere in the methods or results and not defined under the table as a footnote. Other abbreviations should all be defined under the table.	Thank you for your feedback, we add information on SE (standard error) in each table
What was the basis for the classification of the number of visits in Table 1?	Thank you for your comment. We classified the visits originally as those attending just 1 time (1 visits) and those attending more than 1. We further classified those visiting more than 1 by to provide more information on the follow ups visit by the POSBINDU participants.
Minor comments Define POSBINDU in the abstract Non-Communicable should be "Non-communicable" Page 2, line 22: These "figures" should be "This figure" Page 2, line 26; Hypertension "is accounted for" should be "accounts for" "low-middle income countries": were you referring to "low- and middle-income countries?"	Thank you for your detail feedback. We made adjustment to address the comments. Line 15-16 "POSBINDU, a community based activity focusing on screening of Non-communicable diseases (NCDs), mainly hypertension and diabetes, in Indonesia." Line 55 "This figure is" Line 59 "hypertension accounts for" We did refer to low- and middle-income countries, and hence, we made adjustment to the text. "for low and middle income countries"
Reviewer: 3 Dr. Riana Rahmawati, Islamic University of Indonesia Comments to the Author: Thanks for the opportunity to review this manuscript. Generally, this manuscript is well written, well-structured and interesting to read.	Thank you for your feedback which improved our manuscript.
However, I have some concerns regarding the data presentation as detailed in the following: Title: What is the English term for POSBINDU? In the title, the authors used "Integrated Screening Post," but on Page 2 (line 33) they used Integrated Health Post.	Thank you for the detailed feedback. We made changes in the document, and make the term consistent throughout the manuscript, as "Integrated Health Post"
In the Methods (Setting), it is said, "POSBINDU aims to empower communities in screening for NCDs and the risk factors, targeting individuals above 15 years old, particularly those of productive age.[19,20] The main activities	Thank you for your detail and insightful feedback. In the methods section we provide general description

include screening for NCDs (mainly hypertension and diabetes) and the risk factors (i.e., smoking, diet, physical activity, obesity)". These statements reflect the role of POSBINDU in the screening of NCDs' risk factors, including hypertension. However, the authors highlighted "missed opportunity in hypertension screening and risk factors" throughout the paper. What the "risk factors" refer to? Are they CVD or hypertension risk factors? We know that some CVD risk factors are also factors affecting hypertension. The authors, therefore, should carefully present and highlight the main findings.	of POSBINDU activities. However, for the manuscript, we focus mainly on hypertension, which is one of the main NCDs reported in Indonesia, with relatively high underdiagnosis. Additional information is added to further clarify the focus of this study. (Line 86-88) "For this paper, we focus on POSBINDU implementation in screening of hypertension and its risk factor, particularly, since only 30% of hypertensive patients in Indonesia received formal diagnosis.(Turana et al., 2020)
Data collection. There were 2 in-depth interviews. Does it mean 2 participants being interviewed? Who are they?	Thank you for your detailed feedback. The in-depth interviews were conducted on 2 districts health officials which was not available for FGDs due to scheduling conflicts. We add this information on the methods section for further clarification. Line 113-114 "The two in-depth interviews were conducted among health districts department officials."
Outcome (line 36-37). According to the study objective, the primary outcome of this study is POSBINDU implementation instead of missed opportunities in hypertension screening	Thank you for your feedback. One of the main methods to screen for hypertension in communities in Indonesia is through POSBINDU implementation. Hence, we measured the missed opportunities by describing the implementation of POSBINDU. However, we agree with your point of view, and also make some adjustment throughout the paper. We include risk factors for hypertension in the title, aim, and conclusion of the study. In addition, we also made several changes throughout. (Line 1, 31, 78, 281, 311, 374)
Table 1 shows that 23,053 out of 54,224 (around 60%) of participants came from urban North Sumatera. The number of Posbindu in this	Thank you, based on your feedback we re-

region is the highest among other cities. Please briefly explain in the Methods section.	analyze several indicators as well as the proportion of visitors and visits of POSBINDU in our datasets. We found mistakes in our initial analyses, which we have altered in the table (Table 1). The correct data are as follows: 10999 from urban north Sumatra, and 23053 from rural east Java. We added more information in the methods section, explaining this proportion. Line 106-107 “Due to the different number of POSBINDU within each district or PHCs, the number of POSBINDU visitors as well as visits varies by the areas.”
This study revealed a high proportion of missing information on personal and family history. Please briefly explain the kinds of information gathered for these variables.	Thank you for your feedback. We add information on personal and family history in the methods section. (Line 128-132) “Personal and family history of NCDs were also obtained, which include seven (7) diseases: hypertension, diabetes, heart disease, stroke, asthma, cancer, and high blood cholesterol. Complete personal/family history variables were coded 1 if all information was available and coded 0 if at least one of the disease histories was missing. Any personal/family history variables were coded 1 if at least one of the disease histories was available and coded 0 if all of the history information was missing.”
- Table 2. (Risk factor screening). I assume the data report the NCD risk factors instead of hypertension risk factors. Why did the authors include two factors only (obesity and hypertension). What about diabetes and high cholesterol?	Thank you for your suggestion and feedback. This manuscript focus on hypertension and its risk factors. However, data on blood cholesterol had very high missing values. Hence, we did not include this in further analyses i.e. not presenting the proportion of people with high cholesterol, due to the lack of data (only

	15% of samples had information on cholesterol level), and only present hypertension and obesity (more than 75% had information on these variables). We add additional information on this, as well as the number of people entered into the hypertension and obesity analyses in the table, to further clarify the information. (Line 146-149) “Analyses were conducted on missing information, reflecting whether specific procedure in POSBINDU were carried out and reported. Further analyses on proportion of hypertension and BMI status were also conducted. The two indicators were reported due to relatively high availability of these data (92% and 76%) compared to other indicators.”
Page 7, line 43. socialization for this (hypertension screening)..... (comment: hypertension or NCD screening?)	Thank you for your insightful feedback. We reviewed the transcript and notes from the FGDs, and altered the context to POSBINDU. (Line 213) “...Socialization for this (POSBINDU) is needed,”
Page 5 line 57: this paragraph explains the relatively high missing information for hypertension screening. On page 6 (line 3), however, it is said, “we found the highest proportion of available data for blood pressure measurements in all the seven districts. These are contradictory statements. It would be fine if the first sentence changed into “..high missing information for NCD risk factors’ screening”. As mentioned in the Abstract, this study aimed to assess the implementation and contextual barriers of POSBINDU in Indonesia.	Thank you, for the insightful feedback. We agree that the paper focus not only on hypertension, but also its risk factor. Hence, we made adjustment to also include risk factors. This include adjustment in title, as well as within the manuscript (Line 1, 31, 78, 281, 311, 374)
The first sentence of the Discussion also says “In this study, we revealed missed opportunities in input, activities, and output of POSBINDU Implementation” (page 8 line 46). In the Discussion, the authors have explained well the contextual barriers related to the NCD screening. With regard to hypertension screening, the authors need to highlight the finding that showed almost all visitors had had blood pressure examination. The statement “the relatively high missed opportunity in hypertension screening...” (page 9, line 27) therefore need to be evaluated.	Thank you for the insightful feedback, we change the sentence to emphasize that the missed opportunity was mainly on hypertension risk factors and sociodemographic factors. (Line 311-314) “The relatively high missed opportunity in screening for hypertension risk factors, as

	well as sociodemographic characteristics found in this study, portrays suboptimal implementation of POSBINDU. This can be caused by a lack of recording and reporting (monitoring and evaluation fidelity) or lack of measurement (implementation fidelity).”
Page 9, line 40. What does PANDU PTM stand for?	We included the acronym, as well as the English synonym (Line 304-305) “In the MOH, the PANDU PTM (Pelayanan Terpadu Penyakit Tidak Menular, Integrated Health Services for NCDs) and POSBINDU are regulated under the Directorate for Disease Management “
Conclusion. Again, the statement regarding the missed opportunities for hypertension screening needs to be reworded.	Thank you for your feedback. We rephrase the questions to better reflect the finding of this study. (Line 374-376) “This study showed the suboptimal implementation of POSBINDU implementation. Particularly, the missed opportunity in screening for hypertension risk factors in Indonesia.”

VERSION 2 – REVIEW

REVIEWER	Herman, Bumi Chulalongkorn University, Public Health
REVIEW RETURNED	26-Nov-2021

GENERAL COMMENTS	The author addressed all the questions and limitation. The manuscript has more meaningful information and deemed suitable for publication. However some minor issues should be addressed in the future when different researcher plan to conduct similar studies  1. People who become a health cadre or health volunteer has a bias in elaborating their experience of using health service. They tend to have positive attitude compared to common people. Involving health cadre as the "provider" and the "user" of health service unit is not fully acceptable. Thank you for the author who addressed this issue and added more insights in the text to tackle this issue. 2. it is always important to ensure that research related to clinical data should have an outcome measured with reliable measurement tools and consistent procedure. The data of this study relies on the measurement by health cadres, and mostly not of them are health professional. However, the author addressed this issues already. I hope this article would strengthen the primary care service in
--

	Indonesia, particularly in dealing with NCD
REVIEWER	Rahmawati, Riana Islamic University of Indonesia
REVIEW RETURNED	04-Dec-2021
GENERAL COMMENTS	Thank you. The authors have provided a detailed and thorough response to the comments and queries. The following are minor revisions required to clarify the presentation and interpretation of the data.  - The first paragraph after Table 1. The authors changed the age classification. The text need to be revised accordingly (“ with roughly 50% of participants aged over 50 years old (Table 1)”) - Table 2. The title needs revision. This Table presents missing information observed among Posbindu participants and the characteristics of risk factors found in Posbindu. Therefore, the term “Missed opportunity in hypertension screening” in the Title needs rewording. - The first paragraph after Table 2. Kindly consider revising this paragraph to present the findings in Table 2. The statement “.... we found the highest proportion of available data for blood pressure measurements in all the seven districts” contradicts the previous statement “..... the relatively high missing information for hypertension screening”. In the first sentence, the authors might change the term “high missing information for hypertension screening” to “missing information for screening in POSBINDU”. - I enjoyed reading this interesting paper. I hope you will find the comments useful. Best wishes.